# SR-OOD: Out-of-Distribution Detection via Sample Repairing

## ABSTRACT

Out-of-distribution (OOD) detection is a crucial task for ensuring the reliability and robustness of machine learning models. Recent works have shown that generative models often assign high confidence scores to OOD samples, indicating that they fail to capture the semantic information of the data. To tackle this problem, we take advantage of sample repairing and propose a novel OOD detection framework, namely SR-OOD. Our framework leverages the idea that repairing an OOD sample can reveal its semantic inconsistency with the in-distribution data. Specifically, our framework consists of two components: a sample repairing module and a detection module. The sample repairing module applies erosion to an input sample and uses a generative adversarial network to repair it. The detection module then determines whether the input sample is OOD using a distance metric. Our framework does not require any additional data or label information for detection, making it applicable to various scenarios. We conduct extensive experiments on three image datasets: CIFAR-10, CelebA, and Pokemon, and demonstrate that our approach achieves superior performance over the state-of-the-art generative methods in OOD detection.

## 1 INTRODUCTION

Machine learning models are often trained on a specific data distribution, but may encounter unseen data from different distributions in real-world scenarios. This poses a challenge for the security and reliability of machine learning systems, especially for those that are error-sensitive, such as autonomous driving and medical diagnosis. Out-of-distribution (OOD) detection, which aims to identify whether an input data is from the same distribution as the training data, is an important technique for machine learning.

Generative models have shown great potential for OOD detection (17), as they can capture the characteristics of the training data distribution and reject the inputs that are unlikely to be generated by them. Specifically, probabilistic generative models can use the likelihood p(x) as a criterion to measure how well an input data x fits the model, while implicit generative models can use the reconstruction loss between the input data and the reconstructed data as a criterion to measure how well the model can reproduce the input (28).

Despite the advantages of generative models, they can sometimes misidentify OOD samples as in-distribution with high confidence. According to (17), many deep probabilistic generative models trained on the CIFAR-10 dataset tend to assign a higher likelihood to the SVHN dataset. This phenomenon is also observed in the FashionMNIST and MNIST datasets. To explain this issue, several recent works have claimed that deep generative models focus too much on low-level features instead of high ones (25; 22; 32), therefore significantly reducing the OOD detection performance.

Inspired by this, we propose a novel framework for OOD detection, SR-OOD, which leverages semantic inconsistency between original and repaired samples. Our approach first erodes an input sample and then uses a generative adversarial network to repair it. Next, it computes the semantic inconsistency between the original and repaired samples as a criterion for OOD detection. Figure 1 illustrates the whole process. Our framework is based on the observation that eroding a sample can remove some of its semantic information, and repairing it with a GAN can introduce some noise or artifacts. Therefore, the semantic inconsistency between the original and repaired samples can reflect how likely they are to be OOD. Furthermore, we provide an in-depth understanding of

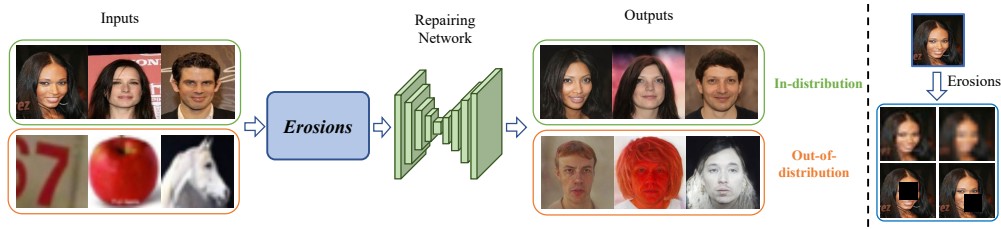

Figure 1: An overview of the proposed SR-OOD framework. The left side shows how SR-OOD distinguishes between in-distribution (in the green box) and OOD (in the orange box) data. Faces are considered in-distribution, while house numbers, apples, and horses are OOD. Inputs are fed independently. The model applies erosion and uses a repairing network specifically trained for this erosion to repair the image. For in-distribution data, the repairing network outputs a similar face. For OOD data, it outputs a face with similar color features but semantically different from the OOD input. The right side displays two erosion methods: downsampling for super-resolution and blacking out for inpainting.

why sample repairing can help OOD detection, as discussion in Section 3.3. We conduct extensive experiments on various datasets and demonstrate that our framework outperforms existing methods in OOD detection.

We present the following contributions in this work:

- We provide an in-depth understanding of why the reconstruction task may cause generative models to focus excessively on low-level features rather than semantics. This is because they rely on the reconstruction and have a tendency to learn the identity map.

- We theoretically analyze why sample repairing can help OOD detection. Based on this, we introduce SR-OOD framework which utilizes repairing techniques to address the OOD detection problem. In the framework, we propose two repairing methods: super-resolution and inpainting. Both methods are simple to implement and computationally efficient.

- As a result, we perform extensive experiments on various datasets to demonstrate the effectiveness of our approaches. The results show that SR-OOD is comparable with existing methods in detecting OOD samples without requiring external data, label information, or time-consuming pipelines.

## 2 PRELIMINARIES

We review the related works in Section A and highlight the main differences and contributions of our work. In this section, we present our hypotheses and claims that motivate our approach.

Our proposed method builds upon several properties of generative models, which have been empirically observed through experiments but cannot be formally proven by theorems. In order to maintain rigor, we formulate these properties as hypotheses and claims.

**Hypothesis 1** *For encoder-decoder based generative models, a well-trained decoder captures the shared features among training data.*

Hypothesis 1 is supported by a wealth of empirical evidence, indicating that a well-trained decoder can consistently generate images with shared features from latent vectors sampled from the latent distribution. For example, models such as pixel2style2pixel (psp) (23) and NVAE (27), trained on human faces, have demonstrated the ability to generate high-quality images of human faces from samples drawn from the latent distribution.

Based on our current knowledge, only one extreme scenario may reject Hypothesis 1: embedding an out-of-distribution (OOD) image using the method proposed in (1) such that the resulting latent space sample generates the OOD image. However, this scenario is not of great significance because

it is deliberately constructed, and the latent sample is dissimilar to the distribution from which we generate regular samples.

**Hypothesis 2** *When the decoder receives latent points encoded from OOD inputs, it will generate images that exhibit some shared features with the in-distribution.*

This hypothesis is supported by the results obtained from our model, as demonstrated in Figure 4, where the generated images from OOD inputs still possess distinguishable facial features that are recognizable to humans. Therefore, this visual validation serves as evidence to support Hypothesis 2.

**Hypothesis 3** *The reconstruction task may cause the encoder and decoder (if trainable) to focus excessively on low-level features rather than semantics, as they are required to learn an identical transformation.*

Recent studies have shown that challenging tasks like context prediction (5), inpainting (21), and reconstructing jigsaw-permuted images (19; 2) can improve a deep learning model's semantic understanding, as evidenced by better performance in downstream tasks. These findings lend support to Hypothesis 3, which suggests that **the reconstruction task may not be as effective as other tasks at promoting semantic understanding**, as it can cause the encoder and decoder to focus too heavily on low-level features.

**Claim 1** *The OOD data, which originates from a different domain, should have distinct shared features compared to the in-distribution data.*

This claim is clear, otherwise the OOD data can not be distinguished from the in-distribution data.

**Claim 2** *The shared features of the SVHN dataset have a significant overlap with those of the CIFAR-10 dataset. Besides, the shared features between SVHN and CIFAR-10 are low-level features.*

Nalisnick et al. (17) first reported that most probabilistic generative models trained on CIFAR-10 exhibit a higher likelihood for SVHN than for CIFAR-10. They explained this phenomenon through second-order analysis, suggesting that SVHN is encompassed within CIFAR-10. (32) subsequently demonstrated that only low-level (local) features are shared between CIFAR-10 and SVHN, by comparing the test BPD of a full autoregressive model with that of a local autoregressive model.

## 3 METHODOLOGY

Based on hypotheses, we formally propose the SR-OOD framework in Section 3.1. Then we provide the implementation details and analysis of the SR-OOD framework in Section 3.2 and Section 3.3.

### 3.1 SR-OOD

Our framework consists of two components: a sample repairing module and a detection module. As shown in Figure 1. In the training stage, we randomly sample an erosion operation $T$ from a set of erosion methods $\mathcal{T} = \{T_1, \ldots, T_m\}$ to apply to a training sample. We then employ a repairing network $R_\theta$ to repair the sample. In the test stage, we fix the erosion operation $T$, and use the same repairing network $R_\theta$ to repair the sample. Finally, we determine whether the input sample is OOD using a distance metric.

To train an effective repairing network $R_\theta$, we use a training dataset, denoted by $\{\mathbf{x}_1, \ldots, \mathbf{x}_n\}$, where $n$ is the number of data samples. The loss function is expressed as Equation 1, which is the same as in (23).

$$\mathcal{L}(\theta) := \frac{1}{n} \sum_{i=1}^{n} \lambda_1 \mathcal{L}^{\text{L2}}(\mathbf{x}_i, \theta) + \lambda_2 \mathcal{L}^{\text{LPIPS}}(\mathbf{x}_i, \theta), \tag{1}$$

where $\lambda_1$ and $\lambda_2$ are two hyperparameters that control the importance of the two losses, i.e., $\mathcal{L}^{\text{L2}}$ and $\mathcal{L}^{\text{LPIPS}}(\mathbf{x}, \theta)$. In specific, $\mathcal{L}^{\text{L2}}$ is the L2 loss that measures the difference between the original input image $\mathbf{x}$ and the repaired image $R_\theta(T(\mathbf{x}))$, as

$$\mathcal{L}^{\text{L2}}(\mathbf{x}, \theta) := \|R_\theta(T(\mathbf{x})) - \mathbf{x}\|_2^2,$$

---

**Algorithm 1** Training stage of SR-OOD

---

**Input:** Training dataset $\{\mathbf{x}_i\}_{i=1}^n$, pre-trained perception network $\phi$, number of iteration $N_{\text{iter}}$, batch size $B$, learning rate $\eta$, repairing network $R_\theta$, loss weight $\lambda_1$ and $\lambda_2$, a set of erosion methods $\mathcal{T} = \{T_1, \ldots, T_m\}$ and its corresponding repairing network $R_\theta$.
**Output:** Trained parameters $\theta$ of the repairing network $R_\theta$.
1: Initialize $\theta$
2: **for** $t$ in $\{1, \ldots, N_{\text{iter}}\}$ **do**
3:    Sample a batch index set $\mathcal{I}$ where $|\mathcal{I}| = B$
4:    **for** $i$ in $\mathcal{I}$ **do**
5:       $u \sim \mathcal{U}(1, |\mathcal{T}|)$                    ▷ Sample $u$ uniformly.
6:       $\mathcal{L}^{\text{L2}}(\mathbf{x}_i, \theta) = \|R_\theta(T_u(\mathbf{x}_i)) - \mathbf{x}_i\|_2^2$
7:       $\mathcal{L}^{\text{LPIPS}}(\mathbf{x}_i, \theta) = \|\phi(R_\theta(T_u(\mathbf{x}_i))) - \phi(\mathbf{x}_i)\|_2^2$
8:       $\mathcal{L}(\mathbf{x}_i, \theta) = \lambda_1 \mathcal{L}^{\text{L2}}(\mathbf{x}_i, \theta) + \lambda_2 \mathcal{L}^{\text{LPIPS}}(\mathbf{x}_i, \theta)$
9:    **end for**
10:    $\theta = \theta - \frac{\eta}{B} \sum_{i \in \mathcal{I}} \nabla_\theta \mathcal{L}(\mathbf{x}_i, \theta)$
11: **end for**

---

**Algorithm 2** Test Stage of SR-OOD

---

**Input:** A sample $\mathbf{x}^*$, pre-trained perception network $\phi$, a threshold $\epsilon$, a selected erosion method $T^* \in \mathcal{T} = \{T_1, \ldots, T_m\}$ and its corresponding repairing network $R_\theta$.
**Output:** A boolean $\delta$ to express whether $\mathbf{x}^*$ comes from OOD.
1: $S(\mathbf{x}^*) = \|\phi(R_\theta(T^*(\mathbf{x}^*))) - \phi(\mathbf{x}^*)\|_2^2$
2: **if** $S(\mathbf{x}^*) > \epsilon$ **then**
3:    $\delta = 1$                    ▷ $\mathbf{x}^*$ is from OOD.
4: **else**
5:    $\delta = 0$                    ▷ $\mathbf{x}^*$ is from in-distribution.
6: **end if**

---

$\mathcal{L}^{\text{LPIPS}}$ is the learned perceptual image patch similarity (LPIPS) (34) loss, as

$$\mathcal{L}^{\text{LPIPS}}(\mathbf{x}, \theta) := \|\phi(R_\theta(T(\mathbf{x}))) - \phi(\mathbf{x})\|_2^2,$$

where $\phi$ is a pre-trained perception network that extracts visual features from images.

In the test stage, we use perception loss $\mathcal{L}^{\text{LPIPS}}$ to determine whether an input image $\mathbf{x}^*$ is OOD. This is based on the intuition that the repaired OOD sample is significantly different from the input sample in terms of visualization, which can be captured by the perception loss. The perception loss is defined as the distance between the feature maps of a pre-trained network, such as VGG or AlexNet, when applied to the input and output images. The efficiency of different loss functions in detecting OOD sample is shown in Table 4. We can see that the perception loss has the highest accuracy among all the losses, indicating that it is more sensitive to the visual discrepancy caused by sample repairing. The training and test processes are illustrated in Algorithm 1 and Algorithm 2, respectively.

**Choices of erosion method**   Each erosion method $T$ can be associated with a corresponding SR-OOD model $(T, R_\theta)$. In particular, when $T$ corresponds to a downsampling operation, the resulting SR-OOD model is referred to as a **super-resolution-based SR-OOD** (SR-OOD$_{\text{SR}}$)[1] model. This is because downsampling is a degradation process that reduces the image resolution, and the goal of super-resolution techniques is to increase the resolution of such degraded images.

On the other hand, when $T$ involves blacking out a rectangular area of the image, the corresponding SR-OOD model is referred to as an **inpainting-based SR-OOD** (SR-OOD$_{\text{inpaint}}$) model. Inpainting refers to the process of filling in the missing parts of an image, and this is the objective when dealing with blacked-out areas in an image.

In Figure 1, we provide illustrations of both super-resolution-based SR-OOD and inpainting-based SR-OOD. It is worth noting that when the erosion operation $T$ corresponds to an identity map,

---

[1]Note that the first "SR" stands for sample repairing, while the second "SR" stands for super-resolution.

the sample repairing task is reduced to a reconstruction task, which we refer to as **reconstruction-based SR-OOD** (SR-OOD$_{Rec}$). However, according to Hypothesis 3, in such cases, the model $R_\theta$ may excessively focus on low-level features rather than high-level semantics, which could result in reduced performance in detecting OOD samples.

## 3.2 IMPLEMENTATION

We choose the pixel2style2pixel (psp) (23) model for our repairing network $R_\theta$. To enhance its performance, we introduce a set of erosion maps that match the type of the given erosion map $T$. Let $\mathcal{T} = \{T_1, \ldots, T_m\}$ denote the set of erosion maps, where $m$ is the total number of erosion maps in $\mathcal{T}$. During the training stage, we randomly sample a $u$ from a discrete uniform distribution $\mathcal{U}(1, |\mathcal{T}|)$ each time we encounter an input image $\mathbf{x}$. We then apply the corresponding erosion map $T_u$ from $\mathcal{T}$ to $\mathbf{x}$. We select $T^*$ from $\mathcal{T}$ by evaluating on a separate validation set using AUROC.

In super-resolution-based SR-OOD (SR-OOD$_{SR}$), $\mathcal{T}$ is the set of downsampling methods that use bicubic interpolation (12) with varying factors. On the other hand, in inpainting-based SR-OOD (SR-OOD$_{Inpaint}$), $\mathcal{T}$ is the set of functions that black out rectangular areas of different sizes and locations in the image. To make it harder for the decoder to reconstruct OOD data accurately, style mixing is applied. Style mixing is a technique that randomly swaps the latent codes of two images at different resolutions, resulting in a mixed image that inherits features from both sources (10). We apply style mixing to each sample using the mean vector of the latent distribution as the source of style. The latent code of ID data is less affected by style mixing than that of OOD data, because the ID data is more similar to the mean vector of the latent distribution than the OOD data. We analyze the effect of style mixing in Section 3.3.

## 3.3 ANALYSIS

To explain our proposed method in detail, we define two mapping functions: $f : X \rightarrow Z$ and $g : Z \rightarrow X$. These functions can map observable data $x$ to a latent vector $z = f(x)$ and reconstruct $x$ from $z$ as $\hat{x} = g(z)$, respectively. Here, $X \in \mathbb{R}^d$ represents the data space, and $Z \in \mathbb{R}^d$ represents the latent space. The generative model aims to learn a target distribution on $x$, denoted as $P_x$, by fitting an approximate model distribution $\hat{P}_x$ to it. The reconstruction error measures the difference between $x$ and $\hat{x}$, as is derived from Osada et al (20):

$$||x - g(f(x))|| \leq \mathrm{Lip}(g) \cdot ||\delta_z|| + ||\delta_x||. \tag{2}$$

where $\delta_z$ and $\delta_x$ represent the numerical errors in the mapping through $f$ and $g$, respectively, and $\mathrm{Lip}(g)$ stands for the Lipschitz constant of the function $g$. The inequality shows that the reconstruction error depends on both the Lipschitz constants of $f$ and $g$, and the numerical errors $\delta_z$ and $\delta_x$. When either $\mathrm{Lip}(f)$ or $\mathrm{Lip}(g)$ is large, the numerical errors $\delta_z$ and $\delta_x$ become large as well, leading to a larger reconstruction error.

The upper bound equation 2 can be used to design algorithms for OOD detection based on the reconstruction error. It can also be used to evaluate and compare different choices of functions $f$ and $g$, such as different neural network architectures or hyperparameters. By measuring or estimating the Lipschitz constants of $f$ and $g$, one can predict how well they can reconstruct in-distribution and OOD data, and choose the best ones accordingly.

In our framework, we consider a scenario where the function $g$ is a pretrained and fixed generator, implying that both $||\delta_x||$ and $\mathrm{Lip}(g)$ are constant. Therefore, we focus on increasing $||\delta_z||$, which requires designing modules in the encoder. We propose two novel techniques to achieve this: sample repairing and style mixing. Our designs can introduce more diversity and variability in the latent space, and make it harder for the decoder to reconstruct OOD data accurately. As a result, it amplify the magnitude of $||\delta_z||$, therefore increasing the overall reconstruction error for OOD samples.

## 4 EXPERIMENTS

This section reports the results of our main experiments, which evaluated the effectiveness of SR-OOD in detecting OOD samples. We also present some ablation studies that investigate the influence of different components in our proposed framework.

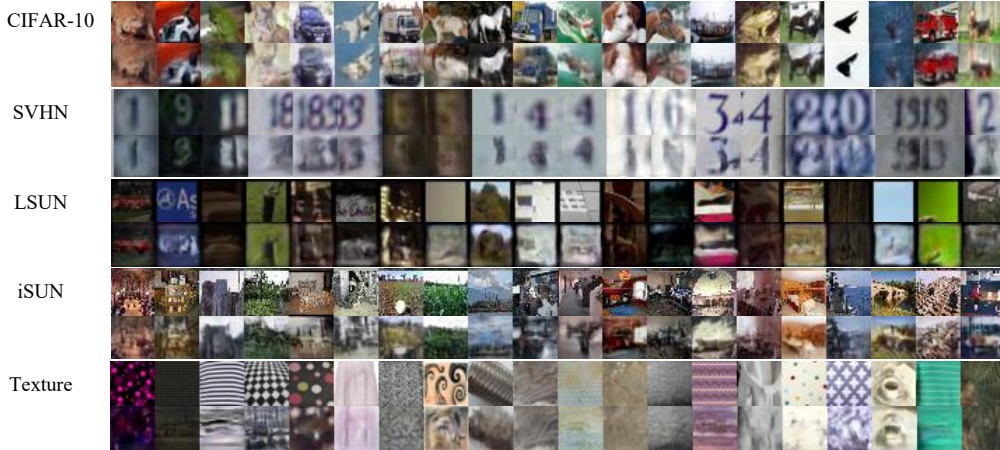

Figure 2: The reconstruction results of SR-OOD$_{Rec}$. The first row shows the original images while the second row shows the reconstructed images. The in-distribution (ID) dataset is CIFAR-10 and all other datasets are out-of-distribution (OOD).

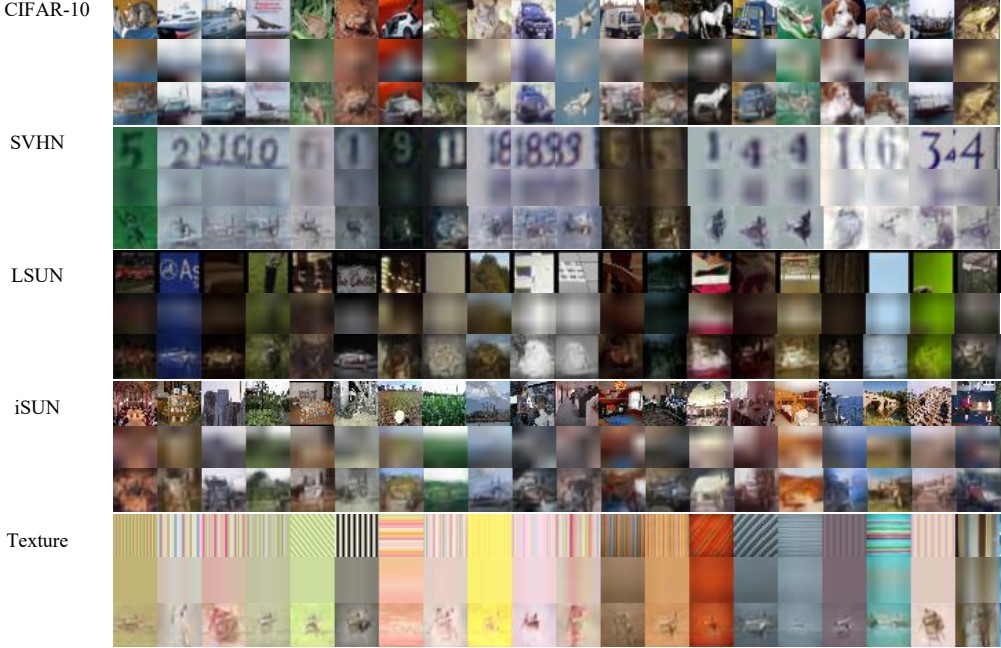

Figure 3: The repaired results of SR-OOD$_{SR}$. The first row shows the original images, the second row shows the images after erosion, and the third row shows the repaired images. The in-distribution (ID) dataset is CIFAR-10 and all other datasets are out-of-distribution (OOD).

## 4.1 MAIN EXPERIMENTS

In our main experiments, we evaluate the performance of several state-of-the-art generative-based OOD detection techniques against three instances (SR-OOD$_{Rec}$, SR-OOD$_{inpaint}$ and SR-OOD$_{SR}$) of our SR-OOD framework for the OOD detection task.

For the OOD detection task, a model is trained on in-distribution training data $\mathcal{D}_{train}^{in}$, then given a single input $\mathbf{x}$, the model outputs a criterion $S(\mathbf{x})$, also known as an OOD score, to indicate the degree to which the input is sampled from OOD. To evaluate the performance of OOD detection methods, we use the practical evaluation introduced in (8): we take an ID test dataset $\mathcal{D}_{test}^{in}$ and an

Table 1: A comparison (AUROC) of OOD detection methods on CIFAR-10 dataset. Higher AUROC values mean better performance. Results are rounded to two decimal places. We implement and report the results of Likelihood Regret and Likelihood on iSUN and Texture datasets, the remaining results are reported directly using the original references. We use a dash "-" to indicate that no results were reported for this method on this particular setting in the original references.

| Method | SVHN | LSUN | iSUN | Texture |
|---|---|---|---|---|
| Glow (diff to None) (25) | 8.80 | 69.30 | - | - |
| Glow (diff to PNG) (25) | 75.40 | 83.60 | - | - |
| Glow (diff to Tiny-Glow) (25) | 93.90 | 89.20 | - | - |
| Glow (diff to Tiny-PCNN) (25) | 16.60 | 16.80 | - | - |
| Likelihood Regret (29) | 87.50 | 69.10 | 37.85 | 43.54 |
| Likelihood (29) | 19.30 | 49.40 | **86.19** | 50.02 |
| SR-OOD$_{Rec}$(ours) | 69.97 | 74.83 | 77.92 | 74.62 |
| SR-OOD$_{Inpaint}$(ours) | 86.37 | 74.26 | 78.27 | 75.88 |
| SR-OOD$_{SR}$(ours) | **94.59** | **95.96** | 84.21 | **92.69** |

Table 2: A comparison of AUROC of our method against baseline methods on CelebA dataset.

| Method | SVHN | CIFAR-10 | CIFAR-100 | VFlip |
|---|---|---|---|---|
| WAIC (3) | 13.90 | 50.70 | 53.50 | 73.40 |
| TT (18) | 98.20 | 63.40 | 67.10 | 60.20 |
| LLR (22) | 2.80 | 32.30 | 35.70 | 60.60 |
| DoSE$_{KDE}$ (16) | 99.30 | 86.10 | 86.70 | 98.30 |
| DoSE$_{SVM}$ (16) | 99.70 | 94.90 | 95.60 | **99.80** |
| SR-OOD$_{Rec}$(ours) | 84.72 | 96.98 | 96.22 | 95.94 |
| SR-OOD$_{Inpaint}$(ours) | 89.17 | 96.69 | 95.66 | 95.75 |
| SR-OOD$_{SR}$(ours) | **99.79** | **99.83** | **99.53** | 90.16 |

OOD test dataset $\mathcal{D}_{test}^{out}$, then for each input $\mathbf{x}^* \in \mathcal{D}_{test}^{in} \cup \mathcal{D}_{test}^{out}$, we calculate its OOD score $S(\mathbf{x}^*)$. We can then evaluate the detector's performance using classic detection metrics such as AUROC, which express how well the criterion $S$ distinguishes between the in-distribution test set and the OOD test set. In the remainder of this subsection, we select CIFAR-10, CelebA, and Pokemon as $\mathcal{D}^{in}$, respectively.

**CIFAR-10 as in-distribution dataset.** In Table 1, the training split of CIFAR-10 serves as $\mathcal{D}_{train}^{in}$, while the test split of CIFAR-10 serves as $\mathcal{D}_{test}^{in}$. In each column of the table, the name of the dataset indicates that it is used as $\mathcal{D}_{test}^{out}$, and the AUROC is calculated using the combined dataset $\mathcal{D}_{test}^{in} \cup \mathcal{D}_{test}^{out}$. For example, SVHN means that SVHN serves as $\mathcal{D}_{test}^{out}$.

We compare our methods with several generative-based OOD detection methods. In Section A.1, we introduce the likelihood ratio method proposed by (25) to detect OOD using external datasets. We denote this method as "Glow (diff to XXX)" in Table 1:

- "Glow (diff to None)" means that we use the likelihood $p(\mathbf{x})$ of Glow, which is trained on the in-distribution dataset $\mathcal{D}_{train}^{in}$, as the OOD score $S(\mathbf{x})$.

- "Glow (diff to PNG / Tiny-Glow / Tiny-PCNN)" means that we use the likelihood ratio between $p(\mathbf{x})$ and a general distribution model as the OOD score. Here, "PNG" refers to the general-purpose image compressor, "Tiny-Glow" refers to a Glow model trained on an external OOD dataset called "80MillionTinyImages", and "Tiny-PCNN" refers to a PixelCNN model trained on the same external dataset.

In Table 1, "Likelihood" refers to the evidence lower bound (ELBO) of the likelihood $p(\mathbf{x})$, as proposed in (29). "Likelihood Regret" is based on a likelihood-ratio method also introduced in (29). SR-OOD$_{Rec}$, SR-OOD$_{Inpaint}$ and SR-OOD$_{SR}$ are our proposed methods introduced in Section 3.1. In the case of CIFAR-10 vs. SVHN, both likelihood-based OOD detection methods, "Likelihood" and "Glow (diff to None)", show small AUROC values, which is consistent with the findings reported by (17).

Table 3: AUROC of the models trained on Pokemon dataset.

| Method | SVHN | CIFAR-10 | CIFAR-100 |
|---|---|---|---|
| MSP (8) | 94.58 | 94.08 | 94.37 |
| MaxLogit (7) | 95.74 | 95.19 | 95.50 |
| SR-OOD$_{Rec}$(ours) | **100.0** | 99.14 | **98.83** |
| SR-OOD$_{Inpaint}$(ours) | 99.64 | **99.36** | 98.41 |
| SR-OOD$_{SR}$(ours) | 99.67 | 99.34 | 98.31 |

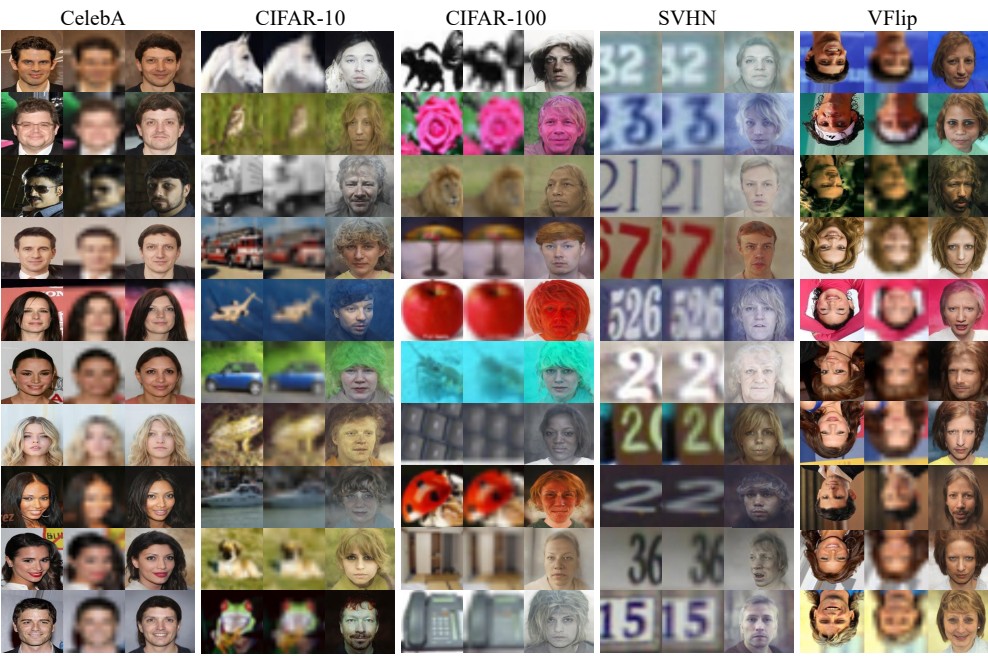

Figure 4: The repaired results of SR-OOD$_{SR}$. The first column shows the original images, the second column shows the images after erosion, and the third column shows the repaired images. The in-distribution (ID) dataset is CelebA and all other datasets are out-of-distribution (OOD).

Our SR-OOD$_{Rec}$ achieves a positive AUROC score ($> 50$) on all of the OOD datasets. However, the AUROC scores of around 70 are not deemed satisfactory. On the other hand, when we changes the reconstruction task to a repairing task, our model SR-OOD$_{inpaint}$ outperforms SR-OOD$_{Rec}$, supporting our Hypothesis 3 that a harder task induces a model that understands more semantics, leading to better OOD detection performance. Among all the methods, our model SR-OOD$_{SR}$ achieved the best performance in 3 out of 4 cases, while in the remaining case (iSUN), SR-OOD$_{SR}$ was not far behind the best-performing model.

**CelebA as in-distribution dataset.** In Table 2, $\mathcal{D}^{in}$ is CelebA. Similar to Table 1, each column stands for a $\mathcal{D}^{out}_{test}$. The "VFlip" refers to the images of CelebA that have been vertically flipped.

For the generative-based OOD detection methods compared in Table 2, "TT" refers to the single-sample typicality test presented in (18), "DoSE" stands for OOD detection using the "density of states" method described in (16), "WAIC" is the Watanabe-Akaike information criterion (3), which uses model ensemble to detect OOD, and "LLR" refers to the likelihood-ratio method. Specifically, we consider the likelihood-ratio method introduced in (22).

We observe that WAIC (3) and "LLR" (22) does not perform well in Table 2. One possible reason for WAIC's poor performance is that the shared features of CelebA are consistent and do not have a significant perturbation in model ensemble. Another possible reason for LLR's poor performance is that CelebA does not provide rich background information, which LLR tries to learn through the model.

The SR-OOD$_{SR}$ achieves the best performance on three out of four OOD datasets, with AUROC values close to 100%. This indicates that our method can effectively distinguish between ID and OOD samples using the super-resolution reconstruction error as the criterion. The DoSE$_{SVM}$ method performs well on all four OOD datasets, especially on the VFlip dataset, where it achieves the highest AUROC value of 99.80%. This suggests that this method can capture the subtle differences between in-distribution and OOD samples using the support vector machine classifier.

**Pokemon as in-distribution dataset.** To further evaluate the scalability of SR-OOD on more diverse dataset, we choose Pokemon dataset (24), which includes data on more than 800 Pokemon from all 7 generations. This dataset contains attributes such as name, type, total stats, HP, and so on. We use this dataset as our ID data and compare the performance of SR-OOD with other methods on different OOD datasets. Due to the training convergence problem of generative-based methods, we only compare our method with two classifier-based methods, MSP (8) and Maxlogit (7).

As shown in Table 3, our method SR-OOD$_{Rec}$ outperforms other methods on three OOD datasets, achieving perfect AUROC values of 100% on SVHN and nearly perfect values of 98.83% on CIFAR-100. The SR-OOD$_{Inpaint}$ and SR-OOD$_{SR}$ methods also show good performance on all three OOD datasets, with AUROC values above 98%. On the other hand, the MSP and MaxLogit methods perform poorly on all three OOD datasets, with AUROC values below 96%. This implies that our method can be applied to diverse dataset like Pokemon, which has more than 800 classes.

**Visualization of reconstruction and sample repairing.** Figure 2 compares the original and reconstructed images of SR-OOD$_{Rec}$ for both CIFAR-10 and OOD datasets. Specifically, for CIFAR-10 and SVHN, We observe that the reconstruction quality for CIFAR-10 is not much higher than that for SVHN dataset. This suggests that SR-OOD$_{Rec}$ is ineffective at distinguishing between these two datasets, as evidenced by its low OOD detection AUC of 69.97 on SVHN.

Figure 3 illustrates the sample repairing process of SR-OOD$_{SR}$ for CIFAR-10 and four OOD datasets. We display the original images, the images after erosion, and the repaired images for each dataset. We observe that the repaired images for CIFAR-10 are much closer to the original images than those for the OOD datasets, which have noticeable artifacts and distortions. This implies that SR-OOD$_{SR}$ can successfully repair the ID samples while degrading the OOD samples. This is consistent with its high OOD detection AUC on the four OOD datasets, as shown in Table 1.

For more reconstructed and repaired results of SR-OOD, please refer to Section C. We also conduct ablation studies to analyze the impact of different components of our approach. The details and results of the ablation studies are given in Section B.

## 5 CONCLUSIONS

In this work, we propose a novel framework for OOD detection called SR-OOD. Our approach uses sample repairing tasks to guide deep generative models to focus more on semantic information and less on low-level features. Typically, deep generative models prioritize low-level features because they attempt to learn an identity map induced by the reconstruction task. To address this issue, we introduce two sample repairing tasks - super-resolution and inpainting - as replacements for the reconstruction task. Our experimental results indicate that our approach effectively detects OOD samples without requiring additional data or label information, which makes our framework practical to apply.

Our framework faces the challenge of handling OOD data that is semantically similar to the in-distribution data, such as adversarial examples. Furthermore, while we used the pixel2style2pixel model as the repairing network in our experiments, any encoder-decoder-based model could be suitable. It would be interesting to explore the use of other models in future research.

The term "semantics" may not be well-defined in our work. However, drawing on previous research in representation learning, we hypothesize that harder tasks may lead to better performance on downstream tasks. In this work, OOD detection can be seen as a downstream task. Improved performance in OOD detection provides evidence to support the idea that the model has a better understanding of semantics. However, further research is needed to understand the specific types of semantics that the model learns.

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

# A   RELATED WORK

## A.1   OOD DETECTION

OOD detection aims to evaluate the ability of neural networks in recognizing inputs that don't lie in the training data's distribution (8). Currently, there are two solutions for detecting OOD data: classifier-based OOD detection and generative-based OOD detection.

**Classifier-based OOD detection:** Dan et al (8) propose a baseline method that assumes the availability of a classifier trained on in-distribution data. When confronted with a new input, this method uses two metrics as the OOD evaluation criterion, i.e., minus max softmax probability (MSP) and information entropy of the categorical distribution. Both metrics perform well because if the classifier is not confident in its prediction, it will suggest that the input comes from an OOD source. For example, if we input an image of cat into a classifier that is trained for detecting dog, the classifier is expected to output a more uniform softmax probability, indicating uncertainty in its prediction.

On the other hand, deep learning classifiers are known to produce uncalibrated softmax probabilities (6), which negatively affects the performance of OOD detection by MSP. Therefore, Liang et al. (14) use the temperature scaling to recalibrate the classifier's probabilities and enhance the performance of OOD detection using MSP. Lee et al. (13) propose to use the entire feature space of the classifier and take the Mahalanobis distance as the OOD criterion. To further enhance the efficiency of OOD detection, Hendrycks et al (9) incorporate OOD data in the fine-tuning stage, driving the softmax probabilities of OOD inputs towards a uniform distribution. Similarly, Malinin and Gales (15) propose a prior network to utilize the OOD data in a Bayesian way. However, these methods have a common limitation: they require label information for training the classifiers, and their performance depends on the quality of the classifiers.

**Generative-based OOD detection:** Nalisnick et al (17) claim that deep probabilistic generative models (PGM), including variational autoencoder (VAE), normalizing flow, and pixelCNN use likelihood as the OOD score. However, these models perform bad when in-distribution is much more complex than OOD, such as CIFAR-10 versus SVHN and FashionMNIST versus MNIST. To address this challenge, several previous works have considered the likelihood-ratio method (31). In specific, Ren et al. (22) utilize the likelihood ratio between the likelihoods of the original PGM and the PGM trained on background data. Schirrmeister et al. (25) employs the likelihood ratio between two PGMs, where one is the original PGM and the other is the PGM trained on an external OOD dataset. Zhang et al. (33) estimates the likelihood ratio proposed in (25) through a binary classifier. Moreover, Zhang et al. (32) adopt the likelihood ratio between a global auto-regressive model and a local auto-regressive model. In addition, Osada et al. (20) propose a new reconstruction error-based approach that employs normalizing flow (NF). By detecting test inputs that lie off the in-distribution manifold, their approach can effectively identify adversarial and OOD examples.

## A.2 GENERATIVE ADVERSARIAL NETWORK

Generative adversarial network (GAN) is a deep learning model that generates synthetic data by alternatively training two neural networks: a generator and a discriminator. The generator produces fake data based on input noise while the discriminator determines whether the input data real or fake. Both networks are trained in a game way until the input of the discriminator are not indistinguishable from the real data. GANs can be used for various applications, such as image synthesis, style editing, and data augmentation. In addition to these applications, the reconstruction error of GANs can be used as criteria for OOD detection.

There have been numerous variants of GANs, each with different characteristics and objectives. One of the most successful ones is StyleGAN (10; 11), which can generate high-quality and diverse images for various domains. It consists of two parts: a mapping network and a synthesis network. The mapping network transforms a random vector from the $\mathcal{Z}$ space into a latent code in the $\mathcal{W}$ space, which is more disentangled and controllable. The synthesis network then uses the latent code to generate the final image. The disentanglement of the $\mathcal{W}$ space enables powerful editing capabilities for images (30; 26; 4). To efficiently perform image repairing tasks, we use the pixel2style2pixel (psp) framework (23), which includes an efficient encoder to encode real images into $\mathcal{W}$ space. It provides a comprehensive range of image-to-image translation capabilities, not only inversion, but also other tasks such as super-resolution, inpainting, and style transfer, which are useful for our purpose.

## B ABLATION STUDIES

**Comparing different distance metrics for OOD detection.** We evaluate the performance of SR-$OOD_{SR}$ on CIFAR-10 using three different loss functions: : LPIPS, L2, and L2+LPIPS, where L2+LPIPS is a combination of LPIPS and L2. Table 4 shows the results. LPIPS achieves the highest AUROC on three out of four datasets (SVHN, LSUN, and Texture), while L2+LPIPS is slightly better on iSUN. L2 has the lowest AUROC on all datasets. This indicates that LPIPS is more effective for most datasets than the other two loss functions, because it reflects the perceptual similarity between images better than pixel-wise distance measures such as L2. Combining L2 and LPIPS does not improve the performance significantly over using LPIPS alone.

Table 4: Comparing the efficiency of different loss functions in detecting OOD samples on CIFAR-10. The results of SR-$OOD_{SR}$ are reported.

| Dataset | L2 | L2+LPIPS | LPIPS |
|---------|-------|----------|---------|
| SVHN | 26.92 | 84.66 | **94.59** |
| LSUN | 70.60 | 94.59 | **95.96** |
| iSUN | 78.54 | **87.35** | 84.21 |
| Texture | 37.97 | 85.77 | **92.69** |

Table 5: The effect of varying the mask offset on the out-of-distribution detection performance of SR-$OOD_{Inpaint}$ on CIFAR-10. The unit of distance is pixels.

| Dataset | offset=8 | offset=4 | offset=0 |
|---------|----------|----------|----------|
| SVHN | 86.25 | 86.30 | **86.37** |
| LSUN | 74.01 | **74.41** | 74.26 |
| iSUN | 78.31 | **78.40** | 78.27 |
| Texture | **75.96** | 75.90 | 75.88 |

**The effectiveness of different mask offsets for OOD detection.** We compare the OOD performance of SR-$OOD_{Inpaint}$ on CIFAR-10 with different mask offsets. The mask offset refers to the distance between the image center and the mask center. This experiment aims to investigate how masking the central or peripheral regions of the image influences OOD detection. Table 5 shows the results. According to the table, changing the offset has a minor impact on out-of-distribution (OOD) performance. This can be attributed to the training process preventing the model from learning an

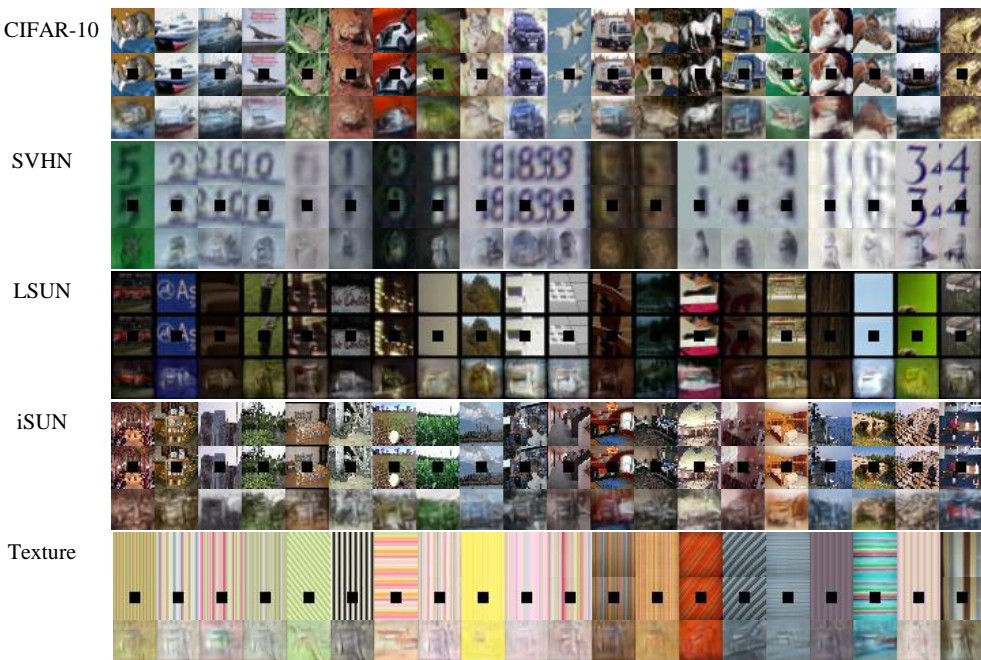

Figure 5: The repaired results of SR-OOD$_{\text{inpaint}}$. The first row shows the original images, the second row shows the images after erosion, and the third row shows the repaired images. The in-distribution (ID) dataset is CIFAR-10 and all other datasets are out-of-distribution (OOD).

identity map. Additionally, the ability to detect OOD is less related to varying the offset during testing.

## C    ADDITIONAL VISUALIZATION OF RECONSTRUCTED AND REPAIRED SAMPLES.

To illustrate more reconstructed and repaired results of SR-OOD, we show the results in Figure 5, Figure 6 and Figure 7. These figures show the reconstructed and repaired results of our method on CIFAR-10 and CelebA images and their corresponding out-of-distribution datasets.

## D    SELECTION OF HYPERPARAMETERS

During the training stage of SR-OOD, we train a repairing network $R_\theta$ for a given erosion method $T$. We use a training set $\{\mathbf{x}_1, \ldots, \mathbf{x}_n\}$ and the loss function:

$$\mathcal{L}(\theta) := \frac{1}{n} \sum_{i=1}^{n} \lambda_1 \mathcal{L}^{\text{L2}}(\mathbf{x}_i, \theta) + \lambda_2 \mathcal{L}^{\text{LPIPS}}(\mathbf{x}_i, \theta), \tag{3}$$

In our experiments, we set $\lambda_1 = 1$ and $\lambda_2 = 0.8$.

On the CIFAR-10 dataset, we re-implement and report the results of Likelihood Regret (29) and Likelihood (29) on the iSUN, and Texture datasets. The remaining results are directly reported using the original references. For the training of Likelihood Regret and Likelihood, we train the VAE for 200 epochs using the default hyperparameters specified in the original paper.

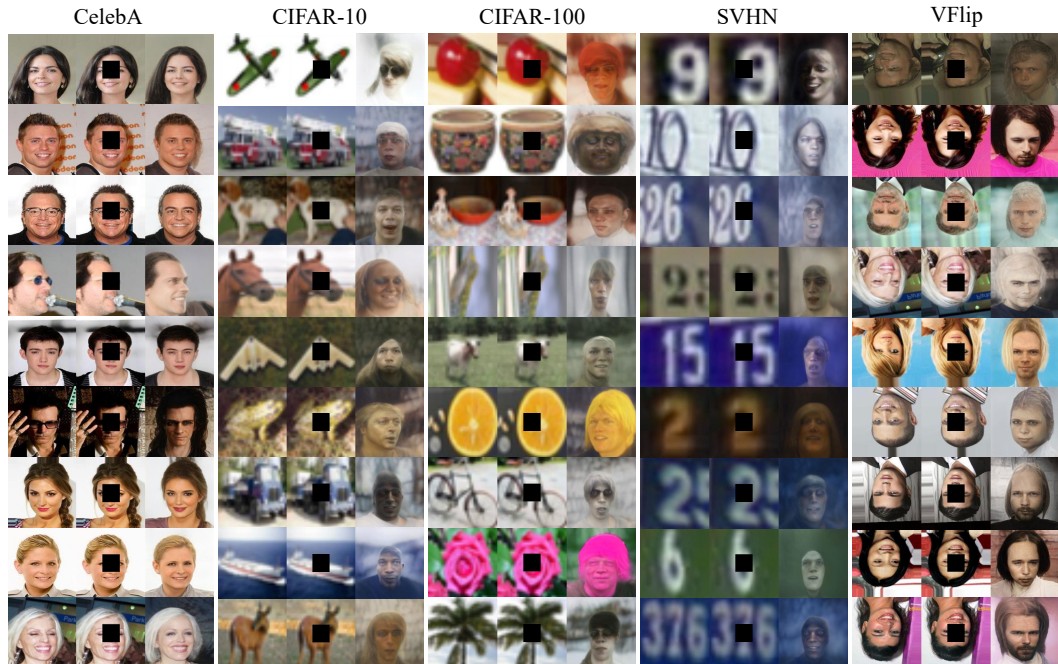

Figure 6: The repaired results of SR-OOD_inpaint. The first column shows the original images, the second column shows the images after erosion, and the third column shows the repaired images. The in-distribution (ID) dataset is CelebA and all other datasets are out-of-distribution (OOD).

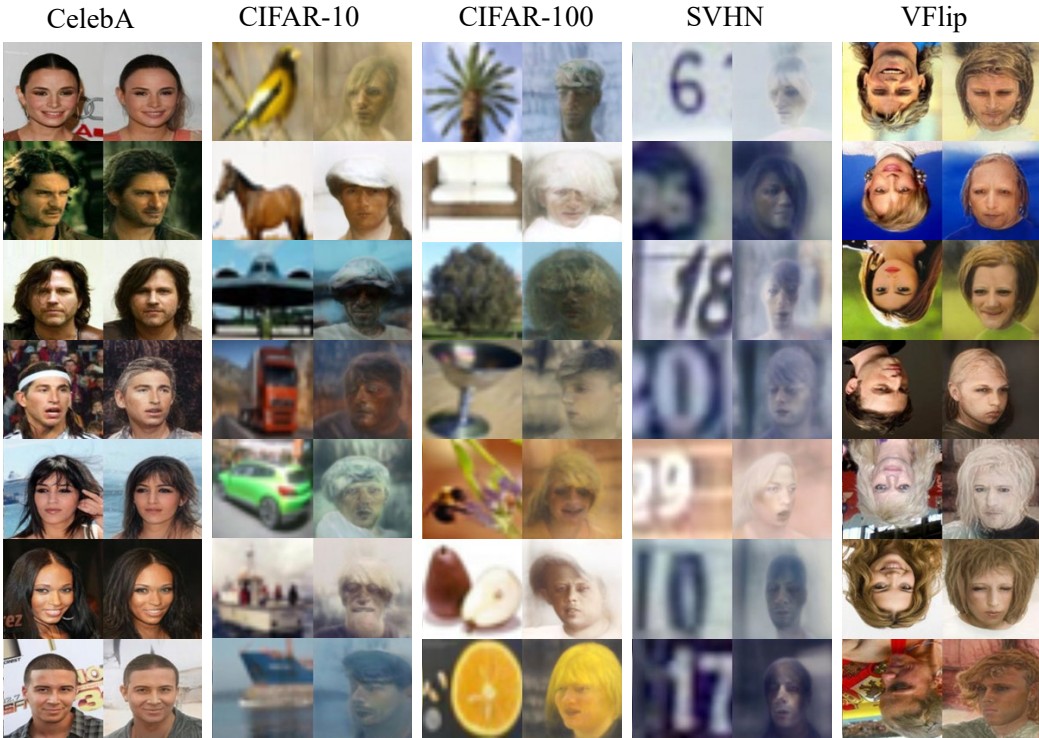

Figure 7: The reconstructed results of SR-OOD_Rec. The first column shows the original images while the second column shows the reconstructed images. The in-distribution (ID) dataset is CelebA and all other datasets are out-of-distribution (OOD).

