# OpenReview forum: "SR-OOD: Out-of-Distribution Detection via Sample Repairing"
_ICLR.cc/2024/Conference — ICLR 2024 Conference Withdrawn Submission_

### Official Review · Reviewer_LA54 · 2023-10-17

**Soundness:** 2 fair
**Presentation:** 2 fair
**Contribution:** 1 poor
**Rating:** 3
**Confidence:** 5

**Summary:**

This paper proposes a sampling repair method for out-of-distribution detection. They train a repairing network to repair the eroded samples. The erosion can be either downsampling or blacking out. Experiments on CIFAR10 and CelebA datasets demonstrate that the method achieves reasonable performance.

**Strengths:**

1. The method intuitively makes sense for far-OOD datasets. In general, I agree with the authors on the hypothesis and claims for far-OOD datasets. The ID and far-OOD datasets can be significantly different so repairing can amplify the semantic inconsistency.

**Weaknesses:**

1. **Usefulness of near-OOD datasets.** As I wrote above, the methodology should work for far-OOD datasets, but how about the near-OOD datasets such as FashionMNIST versus MNIST and CIFAR10 versus CIFAR100? I doubt the efficacy of the methodology for near-OOD datasets as the repairing can be similar.

2. **Scalability to large-scale datasets.** Another concern is that when the training ID dataset scales up to large-scale datasets such as ImageNet-1k,  how would the network repair eroded samples and what is the performance? When the number of ID classes increases, the repairing quality is doubtful and I am not sure whether the repairing part is distinguishable for ID/OOD samples. I believe this problem is also worth further exploration.

3. **Ablation study of L2 and LPIPS loss.** The authors have done some ablation studies using different distance metrics during the inference stage. However, in the training phase, the author uses both L2 and LPIPS loss. It is not clear to me which loss the performance comes from. I suggest the authors conduct more thorough ablation studies.

4. **Recent literature on OOD detection.** The paper only has 34 references but there are many more OOD papers recently. For classification-based OOD detection, there are very strong baselines published on top-tier ML venues [1,2,3,4]. For generative OOD detection, there are also some recent papers which are not taken into account for comparison. I suggest the authors first do a literature review, then compare some of them, and finally conduct a more thorough quantitative evaluation.

>[1] Energy-based out-of-distribution detection. NeurIPS 2020.
>
>[2] React: Out-of-distribution detection with rectified activations. NeurIPS 2021.
>
>[3] RankFeat: Rank-1 Feature Removal for Out-of-distribution Detection. NeurIPS 2022.
>
>[4] Extremely Simple Activation Shaping for Out-of-Distribution Detection. ICLR 2023.
>
>[5] The Tiled Variational Autoencoders: Improving Out-of-distribution Detection. ICLR 2023.
>
>[6] Harnessing Out-of-distribution Examples via Augmenting Content and Styles. ICLR 2023.

5. **Lipschitz analysis.** The analysis of Lipschitz continuity is weak to support the method. If it can really explain the method, the author can use the Jacobian norm of the input as a distance metric. I guess the working mechanism is more relevant to the semantic inconsistency.


Given the above concerns, unfortunately, I can only reject the paper in the current version.

**Questions:**

One minor suggestion is about the format of citing references. Currently, the authors use Parentheses but they are more often used for referring to figures, equations, and tables. Square brackets are used more to improve the readability.

---

### Official Review · Reviewer_RvbJ · 2023-10-29

**Soundness:** 2 fair
**Presentation:** 1 poor
**Contribution:** 2 fair
**Rating:** 3
**Confidence:** 5

**Summary:**

This work, named SR-OOD, proposes another reconstruction-based method for OOD detection. SR-OOD trains a "generative" model (which in fact is a reconstruction model) to repair ID images under certain perturbations (e.g., masking and downsampling). After training, OOD detection can be performed by comparing the reconstruction distance on ID and OOD data. Experiments on CIFAR-10, CelebA, and Pokemon datasets show that SR-OOD can yield improvements over generative-modeling methods and two basic classifier-based methods.

**Strengths:**

The idea is in general intuitive and easy-to-follow.

**Weaknesses:**

### Ambiguity in (theoretical) analysis
1. While the authors clearly state that the hypotheses and claims in Sec. 2 are not formal (and I understand it), I still find many of them are too loose such that clarity is significantly harmed. For example, Claim 1 states that "The OOD data ... should have distinct shared features compared to the in-distribution data." However, in the meantime Claim 2 says "The shared features of the SVHN dataset have a significant overlap with those of the CIFAR-10 dataset", which seems to directly contradicts Claim 1 ("distinct shared features" v.s. "shared features have a significant overlap").

2. I don't see why Claim 2 specifically focuses / restricts itself to the case of SVHN v.s. CIFAR-10. Meanwhile, what's the purpose of presenting the two claims? How do they motivate the proposed method in Sec. 3? I don't see the transition and the logical thinking from claims to the proposed method.

3. The term "reconstruction" should be clearly defined, or at least the authors should clearly illustrate what they mean by "reconstruction". Otherwise, readers (such as me myself) could be confused since the "repairing" method presented in Figure 1 can be thought of as a form of "reconstruction". Only until the last paragraph in Sec. 3.1 did I finally notice that "reconstruction" and "repairing" are two different schemes in the context of this work.

4. I find the theoretical demonstration in Sec. 3.3 fairly weak. First, it is not clear how sample repairing and style mixing can increase $||\delta_z||$. The only explanation, "Our designs can introduce more diversity and variability in the latent space", is ambiguous and not concrete. Second, by increasing $||\delta_z||$ one is only increasing the upper bound (RHS of Eq. 2), which does not guarantee the increase of the actual reconstruction distance (LHS of Eq. 2), which is used in the test stage for OOD detection. Finally, it is unclear how increasing $||\delta_z||$ will affect the reconstruction distance of ID data. This is important since in OOD detection we care about the gap between the score of ID and OOD data. If increasing $||\delta_z||$ enlarges the ID reconstruction error more than OOD, then this design will adversely lead to worse OOD detection performance.

### Experiments
1. One key limitation of the experiments in this work (and in many other OOD detection works), is that ALL OOD datasets considered are far-OOD w.r.t. the ID data (please see the definition and categorization of near-/far-OOD in [1,2]). Long story short, far-OOD samples can exhibit significantly larger low-level differences to ID samples than far-OOD samples (in addition to semantic differences), making them much easier to be detected. For example, when CelebA with human faces is used as the ID dataset, the considered OOD datasets are CIFAR-10/100 (with natural objects), SVHN (with numerical digits), and VFlip (with vertically flipped human faces). What if the OOD data is  human faces of other users (different from those in CelebA)? This is a realistic and intuitively more challenging case. How well can the proposed method perform in this near-OOD setting?

2. The experiments seem to be performed only on small-scale, low-resolution images? I know CIFAR-10/100 are 32x32 images. What is the resolution of CelebA and Pokemon data? Can the proposed method scale to high-resolution images, e.g., 128x128 or even 256x256 ImageNet images?

3. MSP and MaxLogit are two basic methods for classifier-based methods. I would suggest including more SOTA methods such as MDS, KNN, and ASH as identified by [2].

4. The general idea of SR-OOD is to destruct and reconstruct and compare the reconstruction distance. There are several prior works that adopted the same general idea, e.g. [3,4,5]. These works are highly relevant to SR-OOD and should be included as baselines.

### Related work
Similar to my 4-th comment in "Experiments", the related works [3,4,5] are not discussed. In fact, I don't think SR-OOD, which is reconstruction-based, has much to do with generative-modeling-based methods such as Glow and Likelihood (which are used as major baselines and discussed as related works). Generative-modeling methods aim to directly model the likelihood of the input image. Reconstruction-based methods obviously do not have such capability, nor do they perform OOD detection in such way.

### Format
1. The references are being inserted in an absurd way (only have numerical index without author, year information). Also, they seem to be inserted manually and inconsistently: In Appendix A.1 "Classifier-based OOD detection", reference (8) by Dan Hendrycks is cited as "Dan et al", while later reference (9) by the same author is cited as "Hendrycks et al". I believe that the last name should be used, which should be the case if one uses the "\citet" or "\cite" command.

2. Consider using math equation / format for variables and formulations. For example, in the second paragraph in Sec. 1, "... use the likelihood p(x) as a criterion ... an input data x fits the model". Both "p(x)" and "x" should be involved in the math environment for clarity, i.e., "$p(x)$" and "$x$".
-------

[1] Detecting Semantic Anomalies

[2] OpenOOD v1.5: Enhanced Benchmark for Out-of-Distribution Detection

[3] C2AE: Class Conditioned Auto-Encoder for Open-Set Recognition

[4] Classification-Reconstruction Learning for Open-Set Recognition

[5] RaPP: Novelty Detection with Reconstruction along Projection Pathway

**Questions:**

Please see Weaknesses above.

---

### Official Review · Reviewer_9Wxr · 2023-10-30

**Soundness:** 3 good
**Presentation:** 3 good
**Contribution:** 3 good
**Rating:** 6
**Confidence:** 3

**Summary:**

This article investigates the Out-of-distribution (OOD) detection problem and proposes a new framework, SR-OOD, whose main idea is to repair OOD samples in order to reveal semantic inconsistencies between them and in-distribution data. The approach of the article consists of a sample repair module and a detection module. Extensive experiments on several image datasets with visualizations are performed.

**Strengths:**

1.	A framework for OOD detection based on sample repair is proposed, which is a novel and effective idea.
2.	The framework in this paper does not require additional data or labeling information or time-consuming processes, so it can be applied to a variety of scenarios and datasets.
3.	The authors provide some diagrams and pictures to show the effect and principle of their approach, with detailed derivation, principle introduction and rich experiments.

**Weaknesses:**

1.	Not enough details and formulas are given to show how they implement sample repair and OOD detection. For example, in Section 3.1, the authors do not give specifics on how to select the erosion operation T and how to compute the distance metric S(x). In Section 3.2, the authors do not give a specific procedure on how to select the style mixing parameters and how to evaluate the different erosion methods T∗ .
2.	In the experimental section, the authors do not analyze and discuss the experimental results in depth, but simply list some values and pictures. For example, in Section 4.1, the authors do not explain why their method performs better or worse on certain OOD datasets and whether this is consistent with their hypothesis and theory.

**Questions:**

1.	In Section 3.2, could you explain why the pixel2style2pixel model was chosen as the repair network and how it was trained?
2.	In Section 3.2, could you explain why the technique of style blending was used and how the parameters of style blending were chosen?
3.	In Section 3.3, could you give a theoretical analysis of how sample repair can help OOD detection, rather than just providing some intuitive explanation?
4.	In Section 4.1, could you provide comparative experiments with other types of OOD detection methods, such as MSP (8), MaxLogit (7), ODIN (14), etc., be conducted, and the advantages and limitations of their methods in different scenarios be analyzed?
5.	Could you provide the theoretical analysis that sample repair can help OOD detection, including mathematical derivation, theorem proving or experimental verification, etc.?

---

### Official Review · Reviewer_wrBi · 2023-11-09

**Soundness:** 3 good
**Presentation:** 3 good
**Contribution:** 2 fair
**Rating:** 3
**Confidence:** 4

**Summary:**

This paper addresses the problem of OOD detection. It is motivated by the finding that previous reconstruction-based OOD methods focus on low-level features while lacking the ability for semantic understanding. The authors propose SR-OOD to repair images to their original one and use the semantic inconsistency as an OOD score; Experiments on several benchmarks have shown performance advantages.

**Strengths:**

- The idea of adding an extra perception network for original and reconstruction samples intuitively makes sense.
- There are different kinds of downgrading transformation e.g, masking, super-resolution, that have been explored for image repairing.

**Weaknesses:**

Novelty:
- My major concern is that this paper explored a very similar idea to an ECCV 2022 paper [1]. This paper also proposes to repair a masked image with a reconstruction network and then use a further perception network to compute the OOD score for semantic inconsistency. The minor difference from my view is that this paper explored super-resolution as a new type of downsampling augmentation. But overall, they are very similar in high-level ideas.

Experiment:
- I am wondering why authors did not conduct experiments on more common OOD benchmarks such as CIFAR100 and ImageNet.
- Also, the comparing methods only consists of reconstruction-based method while other types of recent OOD methods are missing.
- It seems that SR-OOD_{SR} and SR-OOD_{Inpaint} performs differently on different benchmarks. Do we have a histogram of the OOD score on these two types of models? Do you have any insights that which method fits to which kind of datasets?



[1] Out-of-Distribution Detection with Semantic Mismatch under Masking

**Questions:**

Please refer to the weakness.